# Fucoxanthin-Rich Brown Algae Extract Improves Male Reproductive Function on Streptozotocin-Nicotinamide-Induced Diabetic Rat Model

**DOI:** 10.3390/ijms20184485

**Published:** 2019-09-11

**Authors:** Zwe-Ling Kong, Sabri Sudirman, Yu-Chun Hsu, Chieh-Yu Su, Hsiang-Ping Kuo

**Affiliations:** 1Department of Food Science, National Taiwan Ocean University, Keelung 20224, Taiwan; sabrisudirman@unsri.ac.id (S.S.); yujunhsu@gmail.com (Y.-C.H.); jessieshow1125@gmail.com (C.-Y.S.); 2Biotaiwan Foundation, New Taipei City 24886, Taiwan; williamkuococo@hotmail.com

**Keywords:** diabetes, fucoxanthin, hypogonadism, male reproduction, spermatogenesis

## Abstract

Hypogonadism and oxidative stress are occurring commonly in men with diabetes and associated male infertility. This study aimed to investigate the capability of anti-oxidative and anti-inflammatory properties of fucoxanthin as well as to evaluate its protective effects on male reproduction in diabetic rats. The RAW 264.7 macrophage cells were used to evaluate the anti-oxidative and anti-inflammatory activity. Thirty male Sprague-Dawley rats were induced by streptozotocin-nicotinamide for a diabetes model and fed either with three different doses of fucoxanthin (13, 26, and 65 mg/kg) or rosiglitazone (0.571 mg/kg) for four weeks. The fucoxanthin significantly inhibited nitric oxide production and reduced reactive oxygen species level in lipopolysaccharide-induced RAW 264.7 cells. In the animal study, fucoxanthin administration improved insulin resistance, restored sperm motility, decreased abnormal sperm number, and inhibited lipid peroxidation. Moreover, it restored GPR54 and SOCS-3 mRNA expression in the hypothalamus and recovered luteinizing hormone level, as well as the testosterone level. In conclusion, fucoxanthin not only possessed antioxidant and anti-inflammatory properties but also decreased the diabetes signs and symptoms as well as improved spermatogenesis and male reproductive function.

## 1. Introduction

Diabetes mellitus (DM) has been recognized as a metabolic disorder which occurs due to failure of insulin secretion, insulin action, or both. Type-1 diabetes is characterized by autoimmune-mediated pancreatic β-cell and results in the deficiency of insulin, whereas type-2 diabetes is peripheral insulin resistance [1]. These conditions were characterized by an increase in blood glucose or hyperglycemia [2]. Hyperglycemia increases the level of proinflammatory cytokines, such as interleukin (IL)-1β, IL-2, and IL-6 in the serum of DM patients and has been reported by several studies [3,4]. Changes in antioxidant capacity and oxidative stress also have a positive correlation with cell dysfunction and contribute to DM disease [5]. DM disease impairs some organs, such as pancreas, liver, kidney, and testis [6,7]. In human and experimental models, DM also shows the adverse effects on reproductive function including essential hormone impairment of reproduction (i.e., testosterone, luteinizing hormone, follicle-stimulating hormone) or hypogonadism, sperm motility, and sperm deformity in spermatogenesis [8,9].

Modulation of oxidative stress represents an important strategy for the treatment of DM disease [6]. Insulin secretagogues, alpha-glucosidase inhibitors, and thiazolidinedione (i.e., rosiglitazone) have been used for DM treatment. However, some antidiabetic agents possess adverse effects, such as headache, increased prevalence of gastrointestinal, and cardiovascular diseases [10,11]. Therefore, the investigation of novel antidiabetic agents with less adverse effects is a major challenge in the future research through natural products or functional foods.

Fucoxanthin is a natural product that can be extracted from brown seaweeds (i.e., *Undaria pinnatifida, Saccharina japonica, Sargassum fulvellum*) and is rich in xanthophyll content (one form of carotenoids) [12,13]. The ability of fucoxanthin to ameliorate several diseases, such as hypertension, obesity, anti-inflammatory effects, and anti-carcinogenic activity has been reported [14]. Previous studies reported that fucoxanthin attenuated oxidative stress by scavenging reactive oxygen species and increasing superoxide dismutase activity [15,16]. Fucoxanthin-rich extracts from *Sargassum muticum* reduced inflammation and attenuated colitis disease in mice models [17]. A study reported that fucoxanthin extract shows anti-obesity and anti-diabetic effects by improving insulin resistance and reducing adipokine in white adipose tissue, as well as upregulating glucose transporter 4 skeletal muscle [13]. According to these conditions, we hypothesized that fucoxanthin extract from brown algae (*Laminaria japonica*) possesses anti-diabetes effects and also can improve reproductive function in an experimental diabetic model. Therefore, this study aimed to investigate the ameliorative effects of fucoxanthin-rich extract from *Laminaria japonica* on male reproductive function in a diabetic rat model.

## 2. Results

### 2.1. Effects of Fucoxanthin on Cells Viability, Nitric Oxide, and Reactive Oxygen Species Productions in RAW 264.7 Macrophage Cells

The fucoxanthin extract from brown algae showed non-toxicity to RAW 264.7 macrophage cells (Figure 1A). Nitric oxide (NO) release was significantly reduced after treatment with fucoxanthin in a dose-dependent manner on lipopolysaccharide-induced inflammation RAW 264.7 cells (Figure 1B). As shown in Figure 1C,D, fucoxanthin extracts significantly decreased superoxide anion (O_2_^−^) and hydrogen peroxide (H_2_O_2_) productions after 24 h treatment, respectively, especially in high-doses of fucoxanthin.

### 2.2. Effects of Fucoxanthin on Plasma Glucose, Insulin Level, and HOMA-IR

Figure 2 shows the fasting plasma glucose (FPG), 2 h OGTT, and area under the curve (AUC) of the rats after treatment for four weeks. The untreated-diabetic (DM) group indicated a high level of glucose. Whereas, the AUC of the DM group was significantly higher than both the control (C) and fucoxanthin-treated groups. Groups treated with fucoxanthin extracts had significantly reduced glucose levels when compared to the DM group but without significant differences when compared to the positive control (DMR) group. As shown in Table 1, the insulin level of the DM group is also significantly higher than both the control and fucoxanthin-treated groups. However, those groups treated with both fucoxanthin and rosiglitazone showed significantly improved insulin levels and has no significant difference with the control group. Moreover, fucoxanthin-treated groups also significantly reduced homeostatic model assessment of insulin resistance (HOMA-IR) level of diabetes rats.

### 2.3. Effects of Fucoxanthin on Enzymatic Antioxidant and Oxidative Stress Level of Rats

Diabetic condition significantly causes reduction of enzymatic antioxidant levels as observed in the DM group (Table 2). However, after four weeks of treatment with fucoxanthin, it significantly enhanced enzymatic antioxidant activities including catalase, superoxide dismutase (SOD), and glutathione peroxidase (GPx) in rat plasma, especially in high dose administrations. Additionally, fucoxanthin also significantly enhanced SOD activity in rat testis (Table 3).

The level of reactive oxygen species (ROS) of the sperm was indicated by probe 2′,7′-dichlorofluorescein-diacetate (DCFH-DA) staining and nitro blue tetrazolium (NBT) assay. The level of hydrogen peroxide (H_2_O_2_) and superoxide anion (O_2_^−^) production of the DM group was significantly higher than the control and fucoxanthin-treated groups (Figure 3). Both groups treated with fucoxanthin and rosiglitazone significantly reduced the H_2_O_2_ and O_2_^−^ levels.

The level of malondialdehyde (MDA) was measured as the degree of lipid peroxidation progression. As shown in Figure 4, the MDA level of DM group significantly increased in plasma, testis, and sperm of the rats. Treated with fucoxanthin significantly reduced MDA level especially in a dose-dependent manner.

### 2.4. Effects of Fucoxanthin on Proinflammatory Cytokines

The level of TNF-α and IL-6 were measured as proinflammatory cytokines in plasma and testis of the rats (Figure 5). All doses of fucoxanthin treatment significantly reduced TNF-α and IL-6 levels both in plasma and testis of the rats. Whereas, high-doses of fucoxanthin more effectively decreased their levels.

### 2.5. Effects of Fucoxanthin on SOCS-3 mRNA Expression

Figure 6 showed that the mRNA expression of suppressor of cytokine signaling-3 (SOCS-3) was increased in the DM group. However, its level was reduced by fucoxanthin after treatment for four weeks. Whereas, high-dose of fucoxanthin showed more significant effects to reduce its expression.

### 2.6. Effects of Fucoxanthin on Kiss1 and GPR54 mRNA Expression

As shown in Figure 7A, there was no significant difference in relative Kiss1 mRNA expression between control (C), diabetes untreated (DM) group, and fucoxanthin-treated groups. In the case of GPR54 (Kiss1 receptor), the DM group showed a significantly lower level of relative GPR54 mRNA expression when compared to the control group (Figure 7B). However, the treatment with high doses of fucoxanthin extract significantly improved the level of GPR54 expression.

### 2.7. Effect of Fucoxanthin on Plasma Reproductive Hormones

The level of luteinizing (LH) and testosterone hormones were measured as a representative of male reproductive function. As shown in Table 4, the level of LH and testosterone significantly decreased in the DM group. After treated with fucoxanthin extracts for four weeks, both levels of LH and testosterone were significantly improved. Treated with rosiglitazone (DMR) group also successfully restored the levels of LH and testosterone, however there were no significant difference compared to the DM group.

### 2.8. Effects of Fucoxanthin on Sperm Properties and Testicular Histology

Table 5 shows that the streptozotocin-nicotinamide injection decreased both the total count and motility of sperm as well as increased abnormal morphology. However, groups treated with fucoxanthin extract for four weeks successfully improved sperm motility and reduced the sperm abnormal morphology. In the case of sperm motility, the level was significantly higher when compared to the DM group after treatment with a high dose of fucoxanthin extract.

Figure 8A shows the seminiferous tubule morphology in rat testicles after treatment for four weeks with fucoxanthin extract. This figure showed that in the DM group, the seminiferous tubule structure was separated from each other and appeared shrunken when compared to other groups. Histological observation showed that the diameter of seminiferous tubules decreased and the thickness of the basement membrane was also reduced after STZ/NA-induction. Whereas, treated with fucoxanthin extract significantly improved the thickness of seminiferous tubules after treatment for four weeks, as shown in Figure 8B.

## 3. Discussion

In vitro studies showed that the fucoxanthin extract from *Laminaria japonica* has no effect on the proliferation of RAW 264.7 macrophages cells, which was indicated from high cell viability although in the high concentration of fucoxanthin (Figure 1A). Fucoxanthin extract attenuated oxidative stress by reducing nitric oxide (NO), superoxide anion (O_2_^−^), and hydrogen peroxide (H_2_O_2_) production in RAW 264.7 cells stimulated by lipopolysaccharides (Figure 1C,D, respectively). NO can induce oxidative stress and apoptosis in cell studies as it is recognized as a proinflammatory mediator and marker for oxidative stress status [18,19,20]. A previous study reported that fucoxanthin possesses a protective effect on H_2_O_2_-induced oxidative damage in human hepatic cells by inhibiting translocation of factor-erythroid 2-related factor 2 [21]. Additionally, fucoxanthin also showed an anti-apoptotic effect on carbon tetrachloride-induced hepatotoxicity by enhancing heme oxygenase-1 expression [22].

In vivo studies of fucoxanthin against reproductive functions was evaluated by using diabetes male rats. The diabetes condition was successfully induced by streptozotocin-nicotinamide (STZ-NA). A previous study reported that one of the non-insulin-dependent diabetes mellitus (NIDDM) experimental models without obesity was partially pancreatectomized rats and rats subjected to neonatal administration of STZ. This condition was characterized by β cell mass reduction that was considered also occurs in the NIDDM [23]. Whereas, type 2 diabetes, formerly known as NIDDM was a metabolic disorder, which occurs due to the failure of insulin action [24]. The generation of diabetic conditions was confirmed by increasing plasma glucose concentration and high homeostasis model assessment as an index of insulin resistance (HOMA-IR) (Figure 2 and Table 2). These results were supported by a previous study in which STZ-NA increases plasma glucose concentration in rats after four weeks of administration. STZ is well recognized at inducing the damage of pancreatic B-cells, whereas NA partially protected the cells against STZ [23]. The measurement of plasma glucose levels is one biomarker for diabetes diagnosis and HOMA-IR is widely used for insulin resistance (IR) estimation [25]. In this study, the plasma glucose concentration and IR index were successfully reduced after four weeks of treatment with fucoxanthin extract.

Low levels of enzymatic antioxidant (Table 2 and Table 3) and high levels of oxidative stress condition were observed in the untreated diabetes (DM) group (Figure 3). However, fucoxanthin extracts successfully enhanced enzymatic antioxidant levels9 (catalase; superoxide dismutase, SOD; glutathione peroxidase, GPx) and also ameliorated oxidative stress by reducing reactive oxygen species (ROS) and superoxide anion (O_2_^−^) productions in rat sperm (Figure 3). Additionally, fucoxanthin also reduces the malondialdehyde (MDA) level in plasma, testis, and sperm of rats, as shown in Figure 4. MDA has been used for a biomarker of lipid peroxidation product [26]. SOD antioxidant prevents the tissue from producing highly reactive of O_2_^−^ content by converting them to hydrogen peroxide (H_2_O_2_), whereas catalase acts as catalytic decomposition of harmful H_2_O_2_ to oxygen (O_2_) and water (H_2_O). In the absence of catalase or GPx, H_2_O_2_ leads to the production of reactive hydroxyl radicals and makes free radicals to easier for attack the tissues [27]. Based on these results, fucoxanthin has the potential to ameliorate ROS production and acts as a scavenger for free radicals. This condition can also occur due to the inhibition of hyperglycemia by fucoxanthin. Additionally, previous studies also reported that fucoxanthin reduced ROS production by inhibiting nicotinamide adenine dinucleotide phosphate (NADPH) oxidase-4 (NOX-4) activity [28]. Whereas, NOX-4 are known as ROS-generating enzymes which produce ROS [29]. Hyperglycemia has been known to induce oxidative stress and be involved in the pathogenesis of diabetes [3]. Oxidative stress has been positively correlated with male infertility due to sperm dysfunction [30]. Additionally, a high level of ROS also contributed to abnormal spermatozoa formation [31].

Besides increasing the oxidative stress level, hyperglycemia also increases proinflammatory cytokine levels, such as interleukin-6 and tumor necrosis factor (TNF)-α [4,32]. These proinflammatory cytokines also play a role in systemic inflammation and they are related to insulin resistance in diabetes condition [33]. These proinflammatory cytokines have also caused reproductive dysfunction by triggering testicular apoptosis and atrophy [34]. In Figure 5, untreated-diabetes (DM) groups showed higher levels of TNF-α and IL-6 in plasma and testis than control and treated groups. After four weeks of treatment by fucoxanthin, it reduced proinflammatory cytokine levels.

Additionally, the diabetic condition (DM group) showed a high expression of suppressor of cytokine signaling-3 (SOCS-3) (Figure 6). SOCS family has been reported that inhibit transduction of insulin receptor signaling and resulting in insulin resistance. High level of IL-6 and blood glucose also may induce expression of SOCS-3 in the diabetic patient [35]. We hypothesized that the high expression of SOCS-3 also contributes to severe reproductive dysfunction. Treatment with fucoxanthin successfully inhibited its expression.

In this study, low expression of G-protein coupled receptor 54 (GPR54, Kiss1 receptor) was shown in the diabetic (DM) group. Fucoxanthin treatment significantly increases the GPR54 expression in rat hypothalamic, as shown in Figure 7. Complex Kiss1/Kiss1 receptors are used in reproductive function regulation by controlling gonadotropin secretion and are involved in testicular function [36]. Low levels of Kiss1 or Kiss1 receptors might be caused by high levels of proinflammatory cytokines. A previous study reported that TNF-α reduced Kiss1 receptor expression [37]. Additionally, insulin expression also impaired reproductive function by modulating kisspeptin signaling [38].

Hypogonadism associated with diabetes is characterized by a low level of luteinizing hormone (LH) and testosterone observed in the diabetic condition (Table 4). Diabetes has been reported to disrupt LH from circulating, resulting in increased testis resistance against this hormone, leading to a low level of testosterone [39]. Treatment of diabetic rats with fucoxanthin has an ameliorated effect on the level of LH and testosterone. Based on this condition, we hypothesized that fucoxanthin administration successfully protects the gonadotropin hormone LH against oxidative stress and improve the action of Kiss1/Kiss1 receptor in the hypothalamic-pituitary-gonadal (HPG) axis. Kiss1 has been reported as a gonadotropin hormone stimulator by stimulating gonadotropin-releasing hormone (GnRH) in the hypothalamus [40].

Overall, based on the biochemical observation, fucoxanthin administration successfully ameliorated diabetic condition as well as improved the gonadotropin hormone and might improve male reproductive dysfunction. To confirm the improvement effects of fucoxanthin, we have observed the sperm count, motility and morphology. As shown in Table 5, the diabetic condition leads to reducing sperm motility and increasing abnormal morphology. Diabetes has been known to give rise to an abnormality of sperm morphology, whereas, structural and functional changes throughout sperm tail morphogenesis led to a decrease in sperm motility [41]. Treatment with fucoxanthin increases sperm motility. In the case of the seminiferous tubular structure, Figure 8 showed seminiferous tubular degeneration, shrunken, and separation from each other as well as high thickness basement membrane of seminiferous tubulars in the diabetic group. After treatment for four weeks, fucoxanthin successfully ameliorated seminiferous tubular morphology. This condition was supported by a previous study that the basement membrane of the seminiferous tubular and its diameter decreased in diabetic rats [42]. Additionally, continual renewal of spermatogonia and their differentiation into spermatogenic cells are the regulators for male fertility [43]. A previous study reported that in type-1 diabetes patients, they had lower progressive motility of spermatozoa compared to the normal condition [44].

Used in this study as a positive control; rosiglitazone has been demonstrated to enhance insulin sensitivity [45]. Additionally, rosiglitazone also ameliorates oxidative stress and inflammation by reducing MDA level, enhancing SOD activity and total antioxidant capacity, as well as decreasing TNF-α and IL-6 level in the diabetic patient [46]. On the other hand, rosiglitazone has been reported to associate with cardiovascular diseases [47,48]. Although rosiglitazone was reported as having a low risk of gastrointestinal side effects [49].

## 4. Materials and Methods

### 4.1. Materials

The fucoxanthin was extracted from brown algae *Laminaria japonica*. Fucoxanthin extract was provided by Oryza Oil & Fat Chemical Co., Ltd. (Tokyo, Japan). The RAW 264.7 macrophage cells were supplied from American Type Culture Collection (ATCC). Dulbecco’s Modified Eagle Medium (DMEM) was purchased from GIBCO (Carlsbad, CA, USA). The fetal calf serum (FCS) was purchased from HyClone (Carlsbad, CA, USA). The 3-(4,5-dimethylthiazol-2-yl)-2,5-diphenyl tetrazolium bromide (MTT) powder, nitro blue tetrazolium (NBT) powder, dimethyl sulfoxide (DMSO), 4-(2-hydroxyethyl)-1-piperazineethanesulfonic acid (HEPES) buffer, dichloro-dihydro-fluorescein diacetate (DCFH-DA), and Trypan Blue were purchased from Sigma Aldrich (St. Louis, MO, USA). The glucose enzymatic kit was purchased from Kyokuto Pharmaceutical Industrial Co., Ltd. (Tokyo, Japan) and insulin ELISA kits was purchased from Mercodia AB Inc. (Uppsala, Sweden). Luteinizing hormone (LH) RIA kits were provided by Dr. A. F. Parlow, National Institute of Diabetes and Digestive and Kidney Diseases, National Hormone and Peptide Program (Torrance, CA, USA) and testosterone EIA as well as enzymatic antioxidant (catalase, glutathione peroxidase, superoxide dismutase) commercial kits were purchased from Eugene Chen Co., Ltd. (Taipei, Taiwan).

### 4.2. RAW 264.7 Macrophage Cells Study

#### 4.2.1. Cell Culture

The RAW 264.7 macrophages cells (1 × 10^6^ cells/disk) were cultured in 10 mL of complete culture medium (Dulbecco’s modified Eagle’s medium (DMEM) supplemented with 100 U/mL of Penicillin-Streptomycin and 10% fetal calf serum (FCS)) in 75 cm^2^ disk culture. The cells were incubated at 37 °C in an atmosphere of 5% CO_2_. The cells were sub-cultured every 2 days until the cell count was approximately sufficient to be used. The sub-culture was harvested and prepared by scraping methods by using a cell scraper.

#### 4.2.2. Cells Viability and Nitric Oxide Assay

Cell viability was estimated by the 3-[4,5-dimethylthiazol-2yl]-2,5-diphenyl-tetrazolium bromide (MTT) assay by following the previous method [50]. The cells (2 × 10^5^ cells/well) were plated into 96-well plates in complete medium, pre-incubated for 2 h with fucoxanthin, and added with LPS (1 µg/mL) then incubated at 37 °C for 24 h. After the incubation time, the culture supernatant was taken out and put into a new 96-well plate for the nitric oxide (NO) assay and then each well was washed by a phosphate buffer saline (PBS) and added to 100 µL of MTT (5 mg/mL) solution for 4 h in a CO_2_ incubator at 37 °C. Then, the cells were dissolved with 100 μL of dimethyl sulfoxide (DMSO) and shaken in the dark at room temperature for 15 min. Cell viability was estimated by the colorimetric assay of formazan intensity in a plate reader at 570 nm. The NO assay was determined using the Griess reagent as described by previous methods [51]. Briefly, 50 µL of cell culture supernatant was mixed with 50 µL of Griess reagent (Sulfanilamide and N-(1-Naphthyl)-ethylenediamine), the mixture was incubated at room temperature for 10 min and light was avoided, and the absorbance at 540 nm was measured in a microplate reader.

#### 4.2.3. Reactive Oxygen Species (ROS) Assay In Vitro

The reactive oxygen species (ROS) level was evaluated by using dichloro-dihydro-fluorescein diacetate (DCFH-DA) staining and nitro blue tetrazolium (NBT) reduction assay. The hydrogen peroxide (H_2_O_2_) level was measured by using the probe 2′,7′-dichlorofluorescein-diacetate (DCFH-DA) staining method as described by previous methods [52]. Briefly, RAW 264.7 cells (2 × 10^5^ cells/well) were cultured in 12 wells with LPS (1 µg/mL) either untreated or treated with fucoxanthin and incubated at 37 °C for 24 h. Then, 1 mL of cells culture was loaded with the stain DCFH-DA and incubated for 30 min at 37 °C and then centrifuged at 400× *g* for 4 min in the dark. After washing the cells twice with PBS, the conversion of DCFH to dichlorofluorescein (DCF) was observed, which had a green fluorescent (DCF-DA) color that was detected and evaluated between 500 and 530 nm by flow cytometry. Whereas, superoxide anion (O_2_^−^) was determined by using NBT assay according to previous methods [53]. Briefly, RAW264.7 cells (10^6^ cells/well) were cultured with LPS (1 µg/mL) either untreated or treated with fucoxanthin and incubation at 37 °C for 18 h. The supernatant was removed and added with NBT solution (0.1 mg/mL NBT, 5% FCS, 3% DMSO in 10 mL of DMEM) and incubation for 1 h, then centrifuged to remove the supernatant and dissolved in DMSO. The absorbance was measured at 620 nm by using the ELISA reader.

### 4.3. Animal Experiment

Five-weeks-old of male Sprague-Dawley rats (200 ± 10 g) were maintained under standard laboratory conditions at temperature 22–25 °C under 12 h light/dark cycle. The Institutional Animal Care and Use Committee (IACUC Approval No. 101038; Date: 30 May 2012) of the National Taiwan Ocean University reviewed and approved all protocols. Briefly, thirty SD rats were housed individually and provided standard rodent diet (LabDiet 5001) and water ad libitum. After a 1-week acclimatization phase, the rats were divided into 2 main groups (Figure 9). The first group was a control group (*n* = 5), whereas the second group was a diabetic group (*n* = 25). The diabetes model was induced by intraperitoneal administration of streptozotocin (STZ, 65 mg/kg of body weight) followed by nicotinamide (NA, 230 mg/kg) 15 min later according to a previous method [23]. After a 1-week induction of STZ-NA, the oral glucose tolerance test (OGTT) was performed to evaluate diabetes condition. The diabetes rats were divided into equally into 5 groups. The first group was diabetes rats without any treatment (DM) and another group were treated with either one of three different doses of fucoxanthin (DMF1, 13 mg/kg; DMF2, 26 mg/kg; DMF5, 65 mg/kg) or rosiglitazone (DMR, 0.57 mg/kg) for 4 weeks. The OGTT protocol was performed according to a previously used method [54]. The OGTT was carried out by orally administering glucose (2 g/kg) after 12 h of fasting and blood was drawn to measure glucose levels at 0, 30, 60, 90, and 120 min after glucose injection. The rats were euthanized by CO_2_ exposure in an empty chamber. The whole blood and organs (hypothalamus, epididymis, and testis) were collected then kept for future analysis.

### 4.4. Blood Collection, Supernatant Homogenized Tissue, and Sperm Cells Preparation

The whole blood was collected by using a heparinized-syringe to collection tube. Blood was then centrifuge at 1000× *g* for 15 min to separate the plasma (supernatant). The blood plasma was then kept at −80 °C for future analysis [55]. Whereas, supernatant homogenized tissue (hypothalamus and testis) was adapted from the ELISA kit manufacture’s protocol and previous methods [56,57]. Briefly, 100 mg of tissue was suspended in 900 μL of cold PBS and then homogenized by using a micro-tube homogenizer. The tissue preparation then kept in −20 °C overnight and then thawed in room temperature before use. Afterward, it was centrifuged at 5000 rpm for 15 min. The supernatant was collected for future analysis.

Sperm cells were prepared by using the swim-up technique from epididymis tissue according to the previous method [58]. Briefly, the epididymides tissues were cut in the Roswell Park Memorial Institute medium (RPMI), they were shaken at 100 rpm for 10 min to remove seminal plasma, and the pellet was suspended in fresh RPMI medium. The preparation was shaken at 100× *g* for 10 min, incubated for 37 °C in 5% CO_2_ incubator for 30 min, and the sperm cells were collected from the supernatant for future analysis.

### 4.5. Reactive Oxygen Species Analysis

The catalase, glutathione peroxidase (GPx), and superoxide dismutase (SOD) enzymatic antioxidants of plasma and supernatant of homogenized testis tissue were analyzed by using commercial kits. Whereas, malonaldehyde (MDA) was estimated by following the previous method [59]. Plasma or supernatant of homogenized tissue (100 μL) was mixed with 200 μL of a reactive solution (15% (*w*/*v*) trichloroacetic acid in 0.25 N HCl and 0.375% (*w*/*v*) thiobarbituric acid in 0.25 N HCl) and was placed at 100 °C in a water bath for 15 min. After the cooling process, the mixture was added by 300 μL of *n*-butanol, and it was centrifuged at 1500× *g* for 10 min. The clear supernatant was measured at the 532 nm absorbance. For the sperm MDA analysis, 1 mL of sperm (10^6^ cells/mL) was added to 2 mL of reactive solution and then followed the same procedure with plasma or supernatant of tissue analysis.

### 4.6. Glucose, Insulin, Proinflammatory Cytokines, and Hormones Analysis

The blood glucose was determined by using commercial kits. Insulin, tumor necrosis factor (TNF)-α, and interleukin (IL)-6 were analyzed by using enzyme-linked immunosorbent assays (ELISA) kits. The luteinizing hormone (LH) was analyzed by using immunoradiometric assay (RIA) kits. Whereas, the testosterone was analyzed by using ELISA kits. All the procedures were performed according to the manufacturer’s protocols.

### 4.7. Kiss1, GPR54, and SOCS-3 mRNA Expression Analysis

The quantitative of Kiss1, GPR54, and SOCS-3 mRNA expression was determined by using the conventional reverse transcription polymerase chain reaction (PCR) technique. The RNA was prepared by using RNeasy Lipid Tissue Mini Kit (Qiagen, Hilden, Germany) and cDNA was prepared using HiScript I Reverse Transcriptase (Bionovas Biotechnology Co., Ltd., Ontario, CA, USA) according to the manufacturer’s protocols. PCRs were performed in a 30 μL reaction volume containing 3 μL of cDNA, 10 × Taq buffer (containing 15 mM MgCl2), Taq DNA polymerase (5 U/μL), and dNTP mixture (10 mM) in order. The PCR primers specific to each gene was described in Table 6. Then, 2% agarose gel was prepared and each well was loaded to the DNA of the sample and the DNA marker, which was mixed with tracking dye. After electrophoresis, the gel was stained with ethidium bromide (EtBr). After dyeing, the gel was observed by UV light and photographed. The image was quantified and compared with the marker to determine the molecular weight. The data of gene quantification obtained by computer analysis was divided by internal control, then used as a result.

### 4.8. Sperm Count and Morphology Assay

The procedure of sperm count and morphology assay followed the previous method [60,61]. This method studied the sperm shape abnormality in cauda epididymis of the rats. A thousand sperms per animal were screened to the different types of abnormality in one of the cauda epididymis. Five types of abnormalities, such as amorphous, hookless, banana shape, fused and double-headed were evaluated and finally represented as percentage total abnormality.

### 4.9. Testis Histopathological Analysis

The testis of rats was dissected, fixed in 10% neutral buffered formalin, and processed for paraffin embedding, and stained with hematoxylin and eosin (H&E) according to previous methods [62]. The seminiferous tubules structure was observed under an inverted vertical phase-contrast microscope (Olympus CK2, Japan).

### 4.10. Statistical Analysis

All the values were shown as the mean ± standard deviation (S.D.). To determine the significant (*p* < 0.05) differences among all groups using all the parameters, one-way analyses of variance (ANOVA) with Duncan post hoc test were used using SPSS 22.0 (IBM Corp., Armonk, NY, USA).

## 5. Conclusions

Hyperglycemia induces oxidative stress in the diabetic condition and disturbs reproductive function. This was indicated by hypogonadism and a change in sperm and testicular morphology. Fucoxanthin extract from brown algae has been known to possess various functional properties. In this study, fucoxanthin successfully reduces glucose and oxidative level, as well as enhancing enzymatic antioxidant activity. Additionally, fucoxanthin also decreases proinflammatory cytokine levels, kisspeptin receptors and suppresses cytokine signaling 3 mRNA expressions. As the results of these conditions, fucoxanthin successfully restores the gonadotropin hormone and testosterone level, as well as improves male reproduction function by protecting sperm and testicular morphology. Therefore, fucoxanthin has potential to be developed as an anti-diabetic and as a functional food to improve male reproduction with the diabetic condition.

## Figures and Tables

**Figure 1 ijms-20-04485-f001:**
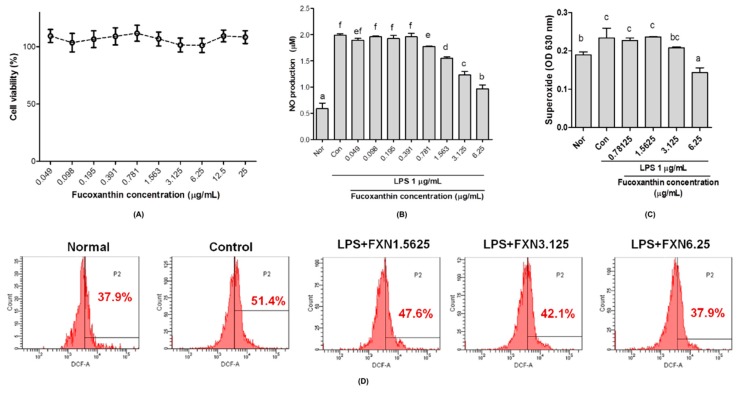
Effects of fucoxanthin (FXN) on (**A**) RAW 264.7 macrophage cells viability, (**B**) nitric oxide, (**C**) superoxide, and (**D**) hydrogen peroxide productions. The cells were stimulated by lipopolysaccharides (LPS, 1 µg/mL). Data are shown as the mean ± S.D. of three independent experiments. The values with different letters (a–f) represent significant differences (*p* < 0.05) as analyzed by Duncan’s multiple range test. Normal, unstimulated cells; control, untreated cells.

**Figure 2 ijms-20-04485-f002:**
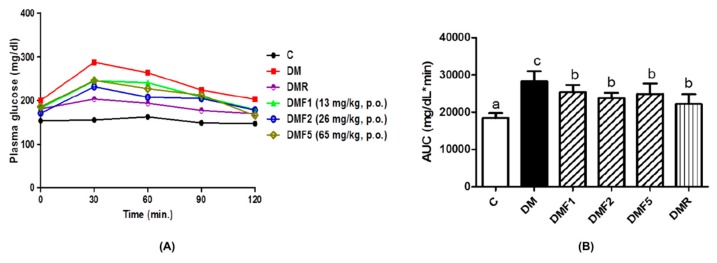
Effects of fucoxanthin on plasma glucose after treatment for four weeks. (**A**) Oral glucose tolerance test (OGTT) and (**B**) area under the curve (AUC) of plasma glucose. Data are shown as the mean ± S.D. (*n* = 5). The values with different letters (a–c) represent significant differences (*p* < 0.05) as analyzed by Duncan’s multiple range test. C, control; DM, diabetes; DMF, diabetes treated with fucoxanthin; DMR, diabetes treated with rosiglitazone.

**Figure 3 ijms-20-04485-f003:**
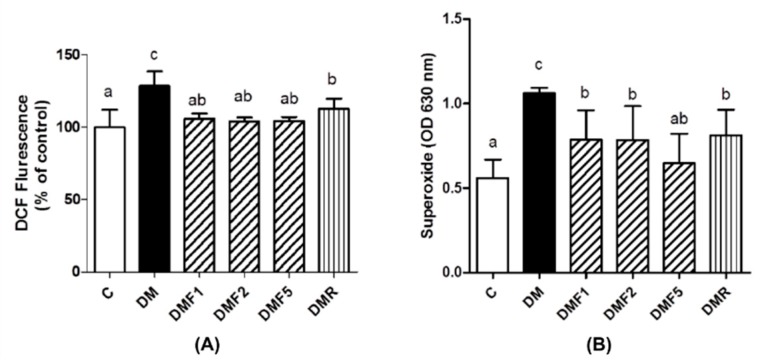
Effects of fucoxanthin on (**A**) reactive oxygen species (ROS) and (**B**) superoxide productions in rat sperm after treatment for four weeks. Data are shown as the mean ± S.D. (*n* = 5). The values with different letters (a–c) represent significant differences (*p* < 0.05) as analyzed by Duncan’s multiple range test. C, control; DM, diabetes; DMF, diabetes treated with fucoxanthin; DMR, diabetes treated with rosiglitazone.

**Figure 4 ijms-20-04485-f004:**
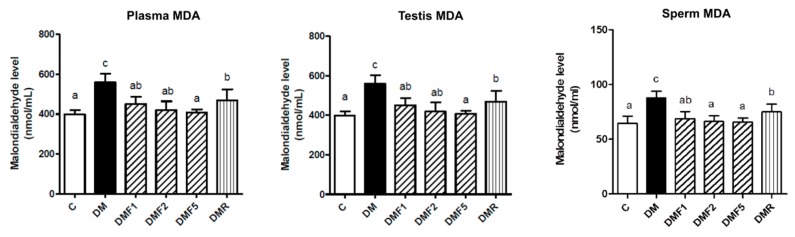
Effects of fucoxanthin on malondialdehyde (MDA) level in rat plasma, testis, and sperm after treatment for four weeks. Data are shown as the mean ± S.D. (*n* = 5). The values with different letters (a–c) represent significant differences (*p* < 0.05) as analyzed by Duncan’s multiple range test. C, control; DM, diabetes; DMF, diabetes treated with fucoxanthin; DMR, diabetes treated with rosiglitazone.

**Figure 5 ijms-20-04485-f005:**
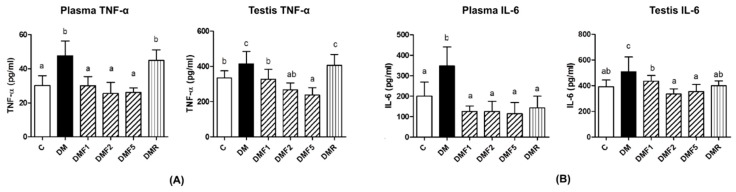
Effects of fucoxanthin on proinflammatory cytokines after treatment for four weeks. (**A**) Tumor necrosis factor (TNF)-α level in plasma and testis, and (**B**) Interleukin (IL)-6 level in plasma and testis. Data are shown as the mean ± S.D. (*n* = 5). The values with different letters (a–c) represent significant differences (*p* < 0.05) as analyzed by Duncan’s multiple range test. C, control; DM, diabetes; DMF, diabetes treated with fucoxanthin; DMR, diabetes treated with rosiglitazone.

**Figure 6 ijms-20-04485-f006:**
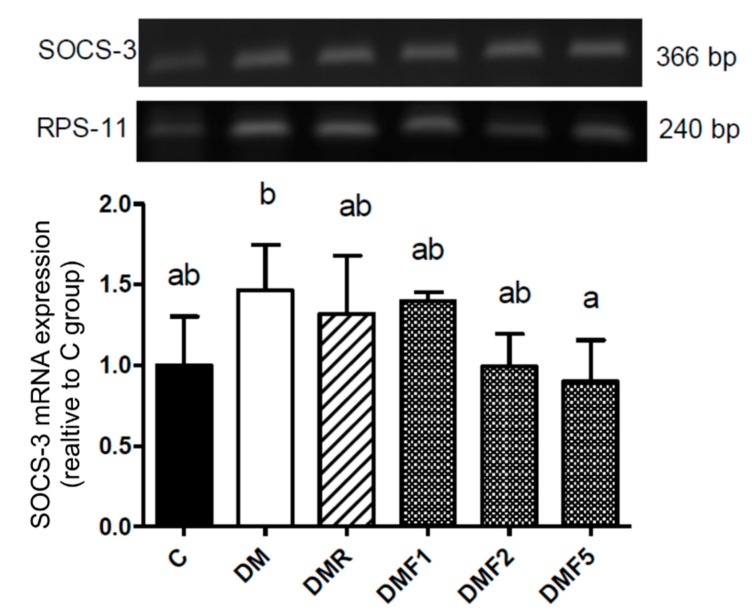
Effects of fucoxanthin on SOCS-3 mRNA expression in rat hypothalamus after treatment for four weeks. Data are showed as the mean ± S.D. (*n* = 5). The values with different letters (a–c) represent significant differences (*p* < 0.05) as analyzed by Duncan’s multiple range test. C, control; DM, diabetes; DMF, diabetes treated with fucoxanthin; DMR, diabetes treated with rosiglitazone; SOCS-3, suppressors of cytokine signaling-3.

**Figure 7 ijms-20-04485-f007:**
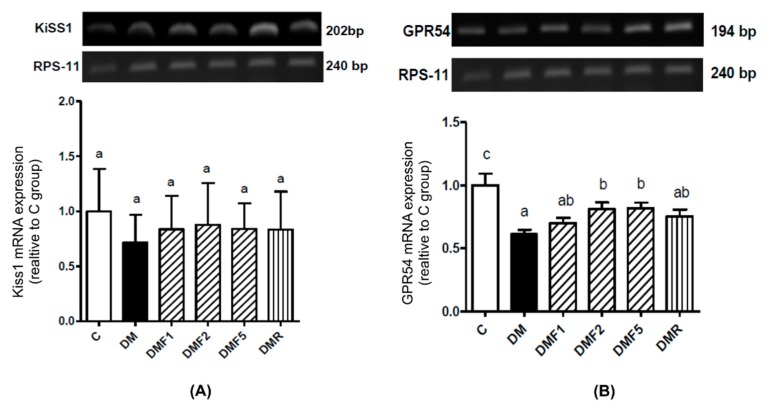
Effects of fucoxanthin on relative (**A**) Kiss1 and (**B**) GPR54 mRNA expression in rat hypothalamus after treatment for four weeks. Data are shown as the mean ± S.D. (*n* = 5). The values with different letters (a–c) represent significant differences (*p* < 0.05) as analyzed by Duncan’s multiple range test. C, control; DM, diabetes; DMF, diabetes treated with fucoxanthin; DMR, diabetes treated with rosiglitazone; GPR54, G-protein coupling receptor (Kiss1 receptor).

**Figure 8 ijms-20-04485-f008:**
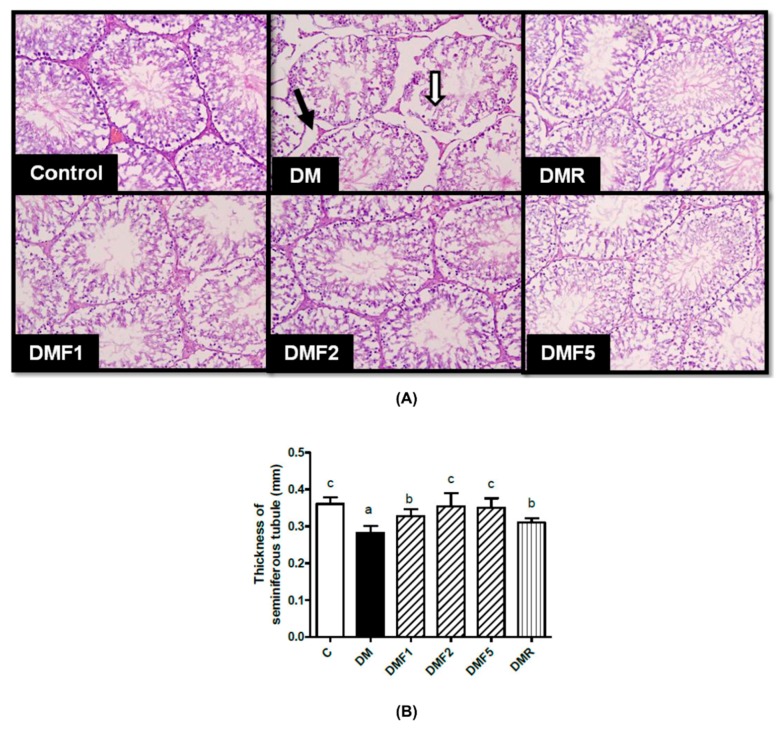
Effects of fucoxanthin on (**A**) testicular morphology and (**B**) thickness of seminiferous tubule diameter after treatment for four weeks. Data are shown as the mean ± S.D. (*n* = 5). The values with different letters (a–c) represent significant differences (*p* < 0.05) as analyzed by Duncan’s multiple range test. Black arrow, Leydig cell; white arrow, Sertoli cell; C, control; DM, diabetes; DMF, diabetes treated with fucoxanthin; DMR, diabetes treated with rosiglitazone.

**Figure 9 ijms-20-04485-f009:**
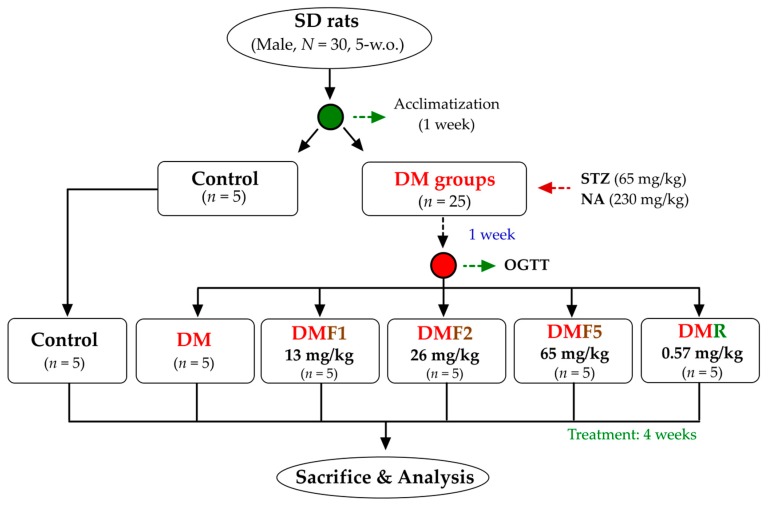
The flowchart of fucoxanthin treatment against streptozotocin-nicotinamide (STZ-NA)-induced diabetes Sprague-Dawley (SD) rat model. DM, diabetes; DMF, diabetes treated with fucoxanthin; DMR, diabetes treated with rosiglitazone.

**Table 1 ijms-20-04485-t001:** Effects of fucoxanthin on fasting plasma glucose (FPG), insulin level, and HOMA-IR after treatment for four weeks.

Properties	C	DM	DMF1	DMF2	DMF5	DMR
FPG (mg/dL)	153.25 ± 20.99 ^a^	211 ± 16.83 ^c^	180.47 ± 10.99 ^b^	174.81 ± 8.52 ^b^	181.55 ± 8.2 ^b^	182.72 ± 13.97 ^b^
Insulin (µg/L)	2.21 ± 0.75 ^a^	4.37 ± 0.63 ^b^	2.81 ± 0.59 ^a^	2.23 ± 0.87 ^a^	2.48 ± 0.78 ^a^	2.46 ± 0.60 ^a^
HOMA-IR	0.82 ± 0.27 ^a^	2.16 ± 0.39 ^b^	1.25 ± 0.19 ^a^	0.95 ± 0.42 ^a^	1.15 ± 0.39 ^a^	1.08 ± 0.18 ^a^

Data are shown as the mean ± S.D. (*n* = 5). The values with different letters (a–c) represent significant differences (*p* < 0.05) as analyzed by Duncan’s multiple range test. C, control; DM, diabetes; DMF, diabetes treated with fucoxanthin; DMR, diabetes treated with rosiglitazone. HOMA-IR: homeostasis model assessment as an index of insulin resistance = (fasting plasma glucose concentration (mmol/L) × fasting plasma insulin concentration (mU/mL)/22.5.

**Table 2 ijms-20-04485-t002:** Effects of fucoxanthin on enzymatic antioxidant activities of rat plasma after treatment for four weeks.

Activities (U/mg Protein)	C	DM	DMF1	DMF2	DMF5	DMR
Catalase	38.72 ± 12.38 ^b^	22.12 ± 4.12 ^a^	29.57 ± 4.89 ^ab^	33.57 ± 9.18 ^b^	38.33 ± 4.55 ^b^	33.06 ± 9.42 ^ab^
SOD	0.56 ± 0.16 ^b^	0.12 ± 0.07 ^a^	0.42 ± 0.20 ^b^	0.40 ± 0.07 ^b^	0.42 ± 0.21 ^b^	0.54 ± 0.13 ^b^
GPx	216.74 ± 44.52 ^cd^	83.95 ± 59.08 ^a^	143.56 ± 43.70 ^ab^	161.83 ± 32.03 ^bc^	191.44 ± 46.18 ^bc^	265.01 ± 58.01 ^d^

Data are shown as the mean ± S.D. (*n* = 5). The values with different letters (a–c) represent significant differences (*p* < 0.05) as analyzed by Duncan’s multiple range test. C, control; DM, diabetes; DMF, diabetes treated with fucoxanthin; DMR, diabetes treated with rosiglitazone; SOD, superoxide dismutase; GPx, glutathione peroxidase.

**Table 3 ijms-20-04485-t003:** Effects of fucoxanthin on enzymatic anti-oxidative activities of rat testes after treatment for four weeks.

Activities (U/mg Protein)	C	DM	DMF1	DMF2	DMF5	DMR
Catalase	105.28 ± 8.59 ^b^	88.38 ± 6.30 ^a^	95.18 ± 8.45 ^ab^	94.36 ± 10.83 ^ab^	99.41 ± 12.68 ^ab^	89.39 ± 4.49 ^a^
SOD	29.71 ± 3.03 ^c^	15.35 ± 3.67 ^a^	16.83 ± 4.42 ^ab^	17.82 ± 2.71 ^ab^	22.78 ± 6.64 ^b^	22.78 ± 4.42 ^b^

Data are shown as the mean ± S.D. (*n* = 5). The values with different letters (a–c) represent significant differences (*p* < 0.05) as analyzed by Duncan’s multiple range test. C, control; DM, diabetes; DMF, diabetes treated with fucoxanthin; DMR, diabetes treated with rosiglitazone; SOD, superoxide dismutase.

**Table 4 ijms-20-04485-t004:** Effect of fucoxanthin on reproductive hormones value of rat plasma after treatment for four weeks.

Hormones (ng/mL)	C	DM	DMF1	DMF2	DMF5	DMR
LH	0.91 ± 0.18 ^b^	0.75 ± 0.03 ^a^	0.96 ± 0.03 ^b^	1.16 ± 0.14 ^c^	1.19 ± 0.10 ^c^	0.85 ± 0.12 ^ab^
Testosterone	1.95 ± 0.04 ^b^	1.02 ± 0.24 ^a^	2.27 ± 0.93 ^b^	2.17 ± 1.03 ^b^	3.21 ± 0.76 ^c^	1.86 ± 0.34 ^ab^

Data are shown as the mean ± S.D. (*n* = 5). The values with different letters (a–c) represent significant differences (*p* < 0.05) as analyzed by Duncan’s multiple range test. C, control; DM, diabetes; DMF, diabetes treated with fucoxanthin; DMR, diabetes treated with rosiglitazone; LH, luteinizing hormone.

**Table 5 ijms-20-04485-t005:** Effects of fucoxanthin on rat sperm properties after treatment for four weeks.

Sperm	C	DM	DMF1	DMF2	DMF5	DMR
Total count (millions)	495.22 ± 76.70 ^a^	405.64 ± 100.01 ^a^	557.60 ± 234.76 ^a^	493.2 ± 144.32 ^a^	540.8 ± 100.16 ^a^	542.00 ± 196.62 ^a^
Motility (% total)	19.23 ± 3.07 ^b^	11.07 ± 5.03 ^a^	15.91 ± 8.07 ^ab^	18.96 ± 6.21 ^b^	18.74 ± 2.39 ^b^	10.95 ± 2.95 ^a^
Abnormal morphology (% total)	8.43 ± 4.04 ^a^	15.76 ± 8.32 ^b^	8.45 ± 3.22 ^a^	10.13 ± 4.07 ^ab^	10.26 ± 1.56 ^ab^	9.88 ± 4.11 ^ab^

Data are shown as the mean ± S.D. (*n* = 5). The values with different letters (a–c) represent significant differences (*p* < 0.05) as analyzed by Duncan’s multiple range test. C, control; DM, diabetes; DMF, diabetes treated with fucoxanthin; DMR, diabetes treated with rosiglitazone.

**Table 6 ijms-20-04485-t006:** The primer sequence and products size for PCR.

mRNA		Primer Sequence	Expected Size (bp)
Kisss1	Sense	5′-TGG CAC CTG TGG TGA ACC CTG AAC-3′	202
	Anti-sense	5′-ATC AGG CGA CTG CGG GTG GCA CAC-3′	
GPR54	Sense	5′-TGT GCA AAT TCG TCA ACT ACA TCC-3′	194
	Anti-sense	5′-AGC ACC GGG GCG GAA ACA GCT GC-3′	
SOCS-3	Sense	5′-CTG GAG CTG CCC GGG CCA GCC-3′	400
	Anti-sense	5′-CAA GGC TGA CCA CAT CTG GG-3′	
RP-S11	Sense	5′-CAT TCA GAC GGA GCG TGC TTA C-3′	240
	Anti-sense	5′-TGC ATC TTC ATC TTC GTC AC-3′

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
