# Peer review of "Fucoxanthin-Rich Brown Algae Extract Improves Male Reproductive Function on Streptozotocin-Nicotinamide-Induced Diabetic Rat Model"

_ijms, 2019, doi:10.3390/ijms20184485_

Round 1

Reviewer 1 Report

Dear authors,

This an interesting paper showing the effect of Brown Algae extract on Diabetes induced early puberty rat (6-7 weeks).  Rat reach puberty at 50 days of age (P50).  This situation in line with type 1 Diabetes that sometimes onset on prepubertal age.  Meanwhile if the author wants to compare with type 2 diabetes, an older rats should be used.

I suggest to explain further on Material Methods

Line 312.  Cell culture instead of Cells culture.   Line 313-316, needs more details on cell culture protocol such as, seeding concentration, flask size for culture, how long the cell reach confluence and harvesting method. Line 348.  This experiment uses 5 weeks old mice with 1 week acclimatisation and 1 week treatment, then culled for data collection.  The rat only at 7 weeks old, and DM induction was started at 5 weeks old.  There is no information how long the diabetic rats receive fucoxathin and rosiglitazone, as a treatment for DM. I suggest (if available) the author add more data on body weight periodically from first day treatment until culling on all groups (control, DM and treated).

Author Response

September 9th, 2019

Dear,

Reviewer #1

Manuscript ID: ijms-591735

Title: Fucoxanthin Rich-Brown Algae Extract Improves Male Reproductive Function of Streptozotocin-Nicotinamide-Induced Diabetic Rat Model.

According to your comments, we have updated our in the manuscript.

This an interesting paper showing the effect of Brown Algae extract on Diabetes induced early puberty rat (6-7 weeks).  Rat reach puberty at 50 days of age (P50).  This situation in line with type 1 Diabetes that sometimes onset on prepubertal age.  Meanwhile if the author wants to compare with type 2 diabetes, an older rats should be used.

Response:

A previous study reported that one of non-insulin-dependent diabetes mellitus (NIDDM) experimental models without obesity was partially pancreatectomized rats and rats subjected to neonatal administration of STZ. This condition was characterized by β cell mass reduction and also consider occurring in NIDDM condition. Whereas, type 2 diabetes formerly known as NIDDM was a metabolic disorder which occurs due to failure of insulin action. à Discussion section (Page 8-9)

I suggest explaining further on Material Methods

Line 312.  Cell culture instead of Cells culture. 

Response:

We have changed it to Cell culture.

Line 313-316, needs more details on cell culture protocol such as, seeding concentration, flask size for culture, how long the cell reach confluence and harvesting method.

Response:

We have updated in this section in our manuscript body. à Cell culture section (Page 10)

Line 348.  This experiment uses 5 weeks old mice with 1-week acclimatization and 1-week treatment, then culled for data collection.  The rat only at 7 weeks old, and DM induction was started at 5 weeks old.  There is no information how long the diabetic rats receive fucoxanthin and rosiglitazone, as a treatment for DM.

Response:

The treatment of diabetic rats with fucoxanthin and rosiglitazone: 4 weeks. à Methods section; Animal Experiment (Page 11)

I suggest (if available) the author add more data on body weight periodically from first day treatment until culling on all groups (control, DM and treated).

Response:

Data not available. All our data have been shown in this manuscript.

Thank you for your valuable comments.

Sincerely,

[Corresponding Author]

Reviewer 2 Report

Authors have elucidated the role Fucoxanthin Rich-Brown Algae Extract in Improving  Male Reproductive Function on Streptozotocin- Nicotinamide-Induced Diabetic Rat Model. Manuscript is very well designed, results are presented clearly. However, i have some minor concerns, after addressing these comments manuscript can be accepted.

I would strongly advise authors to estimate fold induction rather than doing semiquantiative PCR. Results of representative RT-PCR gels does not go along with the graphs showing relative mRNA level.  Authors should also asses the expression of NOX-4, via western blot. It will add more value to the manuscript. As oxidant stress is the focal point of the manuscript.  It would be nice if authors can show some morphological images of ROS generation in animals or in cells. DHE staining of tissues will do the job.

Author Response

September 9th, 2019

Dear,

Reviewer #2

Manuscript ID: ijms-591735

Title: Fucoxanthin Rich-Brown Algae Extract Improves Male Reproductive Function of Streptozotocin-Nicotinamide-Induced Diabetic Rat Model.

According to your comments, we have updated our in the manuscript.

Authors have elucidated the role Fucoxanthin Rich-Brown Algae Extract in Improving  Male Reproductive Function on Streptozotocin- Nicotinamide-Induced Diabetic Rat Model. Manuscript is very well designed, results are presented clearly. However, i have some minor concerns, after addressing these comments manuscript can be accepted.

I would strongly advise authors to estimate fold induction rather than doing semi-quantitative PCR. Results of representative RT-PCR gels does not go along with the graphs showing relative mRNA level.

Response:

We have changed the PCR graphs.

Authors should also asses the expression of NOX-4, via western blot. It will add more value to the manuscript.

Response:

We have no any data related with NOX-4. We have tried to add a reference related with this comment.

Additionally, previous study also reported that fucoxanthin reduced ROS production by inhibiting nicotinamide adenine dinucleotide phosphate (NADPH) oxidase-4 (NOX-4) activity. Whereas, NOX-4 has known as ROS-generating enzymes to produce ROS. à Discussion section (Page 9).

As oxidant stress is the focal point of the manuscript. It would be nice if authors can show some morphological images of ROS generation in animals or in cells. DHE staining of tissues will do the job.

Response:

We have no data about ROS and DHE staining. All our data have been shown in this manuscript.

Thank you for your valuable comments.

Sincerely,

[Corresponding Author]